# Advances in Bioinspired Superhydrophobic Surfaces Made from Silicones: Fabrication and Application

**DOI:** 10.3390/polym15030543

**Published:** 2023-01-20

**Authors:** Zhe Li, Xinsheng Wang, Haoyu Bai, Moyuan Cao

**Affiliations:** 1School of Chemical Engineering and Technology, Tianjin University, Tianjin 300072, China; 2School of Materials Science and Engineering, Smart Sensing Interdisciplinary Science Center, Nankai University, Tianjin 300350, China

**Keywords:** superhydrophobic, silicones, bioinspiration, micro–nano structures, self-cleaning, water repellent

## Abstract

As research on superhydrophobic materials inspired by the self-cleaning and water-repellent properties of plants and animals in nature continues, the superhydrophobic preparation methods and the applications of superhydrophobic surfaces are widely reported. Silicones are preferred for the preparation of superhydrophobic materials because of their inherent hydrophobicity and strong processing ability. In the preparation of superhydrophobic materials, silicones can both form micro-/nano-structures with dehydration condensation and reduce the surface energy of the material surface because of their intrinsic hydrophobicity. The superhydrophobic layers of silicone substrates are characterized by simple and fast reactions, high-temperature resistance, UV resistance, and anti-aging. Although silicone superhydrophobic materials have the disadvantages of relatively low mechanical stability, this can be improved by the rational design of the material structure. Herein, we summarize the superhydrophobic surfaces made from silicone substrates, including the cross-linking processes of silicones through dehydration condensation and hydrosilation, and the surface hydrophobic modification by grafting hydrophobic silicones. The applications of silicone-based superhydrophobic surfaces have been introduced such as self-cleaning, corrosion resistance, oil–water separation, etc. This review article should provide an overview to the bioinspired superhydrophobic surfaces of silicone-based materials, and serve as inspiration for the development of polymer interfaces and colloid science.

## 1. Introduction

Nature is the result of billions of years of evolution and selection, and it has a perfect and diverse ecosystem with thought-provoking natural phenomena. In addition to being attracted to the variations in nature, inspiration ought to be drawn from nature to find better solutions for our progress and development. In recent years, superhydrophobic materials [1], which are often the subject of research, have been inspired by flora and fauna in nature, such as lotus leaves [2], morphology molesta floating leaves [3], butterfly wings [4], fly eyes [5], cicada wings [6], gecko feet [7], shark skin [8], legs of water striders [9], and rose petals [10], as shown in Figure 1. After decades of research by researchers, it has been found that there are various artificial superhydrophobic surfaces, such as superhydrophobic particles (silica powder [11], metal powder [12], metal oxide powder [13], polystyrene powder [14]), superhydrophobic porous materials (membrane [15,16,17], concrete [18], textile [19], sponge [20]), and superhydrophobic surface coatings that can coat the surfaces of various materials (wood [21], aluminum sheets [22], copper sheets [23], and various other substrates [24]). Many methods of preparing superhydrophobic surfaces are achieved with the involvement of silicones. Previously, Zhang [25] counted that about 25% of the papers on superhydrophobicity were based on silanes and silicones, proving the importance of silicones in the preparation of superhydrophobic materials. These advantages come from the properties of silicones: (1) Most silicones possess many hydrophobic groups similar to —C_n_H_2n+1_, —CH=CH_2_, —C_6_H_5_, —X, so silicones are inherently hydrophobic materials. So, organosilicon monomers with multi-hydrolysis functional groups can provide low-surface-energy surfaces or modify the surface of other materials to make them hydrophobic; (2) silicones can easily polymerize through dehydration condensation to form micro-/nano-structures, or act as binders to stabilize other particles on the surface, both of which can generate sufficient surface roughness for superhydrophobic interfaces. (3) Most of the reaction conditions of organosilicon are relatively gentle, and the reaction can be completed under acidic and alkaline conditions, without imposing high temperature and pressure, which makes it simple and fast in the preparation process and convenient for mass production. (4) The main chain of organosilicon products is Si-O, and the double bond is less present, so it is not easy to be decomposed by ultraviolet light and ozone. Similarly, the Si-O bond of the molecule does not break or decompose under high temperature, so the organosilicon have both UV resistance and thermal stability. (5) Silicone has low toxicity, except for a tiny portion containing fluoro-silane, which facilitates mass production and practical applications in our daily life. Therefore, silicones have a significant advantage in the preparation of superhydrophobic materials. However, there are many kinds of silicones and various ways to synthesize and assemble the superhydrophobic interfaces, so this article aims to provide an overview of silicone-based superhydrophobic coatings from fabrication to application.

First, we overview some basic concepts of superhydrophobicity, such as contact angle, rolling angle, dynamic contact angle, the Young-type equation, etc. Then, we introduce the classification and properties of organosilicon monomers and polymerization methods and how they can play a role in the superhydrophobic preparation process; for example, they can reduce the surface energy by grafting, build micro–nano structures by condensation cross-linking, and act as a binder in the reaction. After that, we focus on the preparation methods and application directions of superhydrophobic surfaces on silicone substrates. Finally, we will summarize the challenges faced by silicone superhydrophobic materials in the conclusion section, as shown in Figure 2. Overall, this review should help the reader to understand how silicones prepare superhydrophobic materials and take advantage of them.

## 2. Liquid-Repellent Surfaces and Theory

### 2.1. Young’s Equation

With regard to the solid–liquid contact surface, we call it the “ideal state” if the solid surface is smooth enough to neglect friction and has a uniform chemical composition and is in a perfectly horizontal state. In the ideal state, we can use Young’s equation [26] to determine the intrinsic contact angle. In 1805, Young proposed the famous Young’s equation using hydrophobicity characteristics.
(1)$$\gamma_{\mathrm{SG}}=\gamma_{SL}+\gamma_{LG}\cos\theta_{e}$$

As shown in Equation (1), *θ* is the contact angle of the solid intrinsic, *γ_SG_* is the surface tension of solid–gas, *γ_SL_* is the surface tension of solid–liquid, and *γ_LG_* is the surface tension of gas–liquid.

Through Young’s equation, we find that the contact angle of a solid surface is determined by the surface tension with three boundaries, so we can enhance the hydrophobicity of the solid surface by changing the chemical composition of the solid surface so as to increase the contact angle, and a common method is to modify the solid surface with low-surface-energy substances. Some scientific members found that modification using low-surface-energy substances on smooth materials can increase the contact angle of smooth surfaces up to 120°, which provides a reliable methodological idea for the preparation of superhydrophobic materials.

### 2.2. Contact Angle

The wettability of a solid surface is usually characterized using the contact angle between the droplet and the interface [27]. When a drop is added to a smooth solid surface, the droplet will reach an equilibrium state and will no longer spread; because of the surface tension equilibrium, a minimum of the system energy is reached, thus making the droplet steady or sub-steady. After equilibrium, at the intersection of gas, liquid, and solid phases, the angle formed by the tangent line of the liquid–gas interface and the horizontal line of the solid–liquid interface (including the angle inside the liquid) can be made, and we call this angle the contact angle, as shown by *θ* in Figure 3a.

### 2.3. Dynamic Contact Angle

The static contact angle was mentioned earlier, that is, the contact angle obtained when a liquid droplet is resting on a smooth, stable, not easily deformed, ideal solid surface. However, in reality, the solid surface is different to the ideal state; the microstructure and chemical composition of the surface are not uniform, so the contact angle of the same surface is variable, which means that the contact angle is a range of numerical variation. The upper value of this range is the forward contact angle *θ*_1_, referred to as the forward angle; the lower value is the backward contact angle *θ*_2_, referred to as the backward angle. These two angles are dynamic contact angles; of more practical significance is the contact angle hysteresis, that is, the difference between the forward angle and the backward angle, Δ*θ* = *θ*_1_ − *θ*_2_. Contact angle hysteresis is considered the main obstacle for droplet movement on the surface [28].

The measurement of the advancing and receding angles generally uses the incremental and decremental volume method [29]. The specific operation is to insert a syringe needle into the center of the droplet and inject liquid into the droplet so that the droplet volume increases; along with the increase in droplet volume, the solid–liquid contact area begins to expand, and it can be found that the contact angle does not change significantly, and the contact angle measured at this time is the advancing angle. A similar method is used for the backward angle, where a syringe needle is used to suck up the droplet, and the contact angle measured during the contraction process is the backward angle, as shown in Figure 3b.
Figure 3(**a**) Schematic diagram of contact angle and Young’s equation. (**b**) Schematic diagram of dynamic contact angle measurement. (**c**) Schematic diagram of rolling angle measurement. (**d**) Hydrophobic model and its prototype in nature and electron micrograph. Reprinted with permission from Ref. [30]. 2017, J. Chem. Educ. (**e**) Superhydrophobic model and its prototype in nature and electron micrograph. Reprinted with permission from Ref. [2]. 2011, Beilstein J. Nanotechnol. (**f**) Slippery liquid-infused porous surfaces (SLIPS) model and its prototype in nature and electron micrograph. Reprinted with permission from Ref. [31]. 2016, Nature.
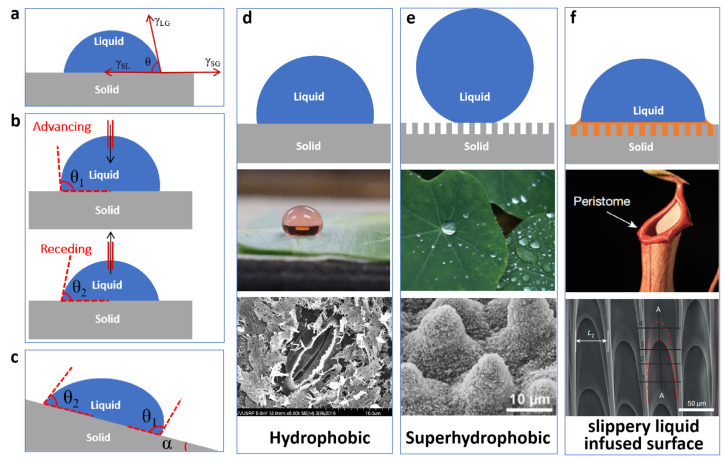


### 2.4. Rolling Angle

The rolling angle is a measure of the adhesion of the object to a certain extent. A drop of liquid on an inclined solid surface will roll down due to gravity, and the smallest angle that can make the drop slide downward is called the rolling angle; as shown in Figure 3c, α is the rolling angle. Our measurement method is the tilted surface method; the droplet on the solid surface can be tilted and slowly change the solid tilt angle. Due to the adhesion of the solid surface droplet, it cannot immediately overcome the gravitational force downward slide, but along with the increase in the angle of gravitational force which also increases to a certain extent, the droplet will slide down, recording the angle α (Figure 3c) in the sliding process; *θ*_1_ and *θ*_2_ are also the forward angle and the backward angle.

### 2.5. Definition of Superhydrophobicity

Generally, when the contact angle between a material and a water droplet is greater than 90° as in Figure 3d, we consider this material to be hydrophobic, such as rubber, plastic, and leaves in nature, including the non-treated collard leaves in the figure [30]. When the contact angle of a material is less than 90°, this material can be called a hydrophilic material. In addition, researchers consider those interfacial materials with contact angles greater than 150° and rolling angles less than 10° to be defined as superhydrophobic materials [32], as in Figure 3e. The most typical superhydrophobic surface in nature is the surface of the lotus leaf. In the schematic diagram of Figure 3e, the droplet is in the Cassie state and there is an air layer between the droplet and the solid due to the high roughness of the superhydrophobic surface and the presence of many gaps; if a lubricant, such as dimethyl silicone oil, is injected into the porous gaps, a slippery liquid-infused porous surface is formed, which is illustrated in Figure 3f. In addition to the possibility of re-injecting lubricants into superhydrophobic surfaces [33], it is also possible to inject lubrication into porous hydrophobic materials [34], which can also form slippery liquid-infused porous surfaces (SLIPS) (The abbreviations appearing in this paper are summarized in the Abbreviations). The biological inspiration for SLIPS is derived from hogweed, a rough surface structure combined with a lubricating fluid.

## 3. Silicones

### 3.1. Silicone Monomer

Silanes can be divided into two main categories based on the atom to which the functional group is said to be attached.

The first category is silanes with the functional group directly attached to the silicon atom, which can be represented by R_n_SiX_4−n_, where R is alkyl, aryl, arylalkyl and hydrogen, etc.; X is a hydrolyzable functional group, such as halogen (main chlorine), alkoxy, acyloxy, amino, and hydrogen, and they are collectively called silicon-functional organosilanes.

The second category is the functional group attached to an organic group other than Si, which can be represented by (YR′)_n_SiX_4−n_, where Y is a functional group (such as NH_2_, OCOCMe=CH_2_, Cl, OH, etc.); R′ is a sub-alkyl group; and X is a hydrolyzable functional group such as halogen, MeO, EtO, AcO, MeOC_2_H_4_O, etc. When n is 1~3, these silanes are collectively referred to as carbon-functional organosilanes.

#### 3.1.1. Organohalogenated Silanes

Chemical formula of organohalogen silane can be represented as R_n_SiX_4−n_ (R is Me, Et, Vi, Pr, Ph, etc.; X is F, Cl, Br, and I; and n = 1~3). The most representative one is organochlorosilane, which is the most important raw material for the preparation of silicone polymers (industrial silicone oil, silicone rubber, and silicone resin) and other silicon-functional silanes.

Si-X bonds are much more active than Si-C and C-X bonds because of their ionic, covalent, and double-bonding properties, and Si-X bonds can be replaced by nucleophilic atoms or groups to produce silanes with corresponding substituents.
$$\begin{array}{c} \;\equiv \mathrm{Si}-X+\mathrm{HOH}\to \;\equiv \mathrm{Si}-\mathrm{HO}+\mathrm{HX}\\ \equiv \mathrm{Si}-X+\mathrm{ROH}\to \;\equiv \mathrm{Si}-\mathrm{OR}+\mathrm{HX}\\ \begin{array}{c} \equiv \mathrm{Si}-X+\mathrm{HOAc}\to \;\equiv \mathrm{Si}-\mathrm{OAc}+\mathrm{HX}\\ \equiv \mathrm{Si}-X+{\mathrm{RNH}}_{2}\to \;\equiv \mathrm{Si}-\mathrm{NHR}+\mathrm{HX} \end{array} \end{array}$$

Organohalosilanes can undergo hydrolysis to form silanols. The rate of the hydrolysis reaction of organohalosilanes accelerates with an increase in polarity and the number of Si-X bonds, but slows down with an increase in site resistance and the number of organic groups. The Si-OH bond can be easily condensed to polysiloxane via dehydration, because silanol is not very stable, especially under the action of real acids.
$$\begin{array}{c} 2R_{3}\mathrm{SiX}+H_{2}O\;\to 2{\;R}_{3}\mathrm{SiOH}\;\overset{-H_{2}O}{\to }{\;R}_{3}{\mathrm{SiOSiR}}_{3}\\ R_{2}{\mathrm{SiH}}_{2}X_{2}+2H_{2}O\;\to R_{2}\mathrm{Si}{\left(\mathrm{OH}\right)}_{2}\;\overset{-H_{2}O}{\to }\;\mathrm{HO}{\left(R_{2}\mathrm{SiO}\right)}_{n}H\;\mathrm{or}\;{\left(R_{2}\mathrm{SiO}\right)}_{n}\\ \begin{array}{c} {\mathrm{RSiX}}_{3}+3H_{2}O\;\to \mathrm{RSi}{\left(\mathrm{OH}\right)}_{3}\;\overset{-H_{2}O}{\to }\;{\left({\mathrm{RSiO}}_{1.5}\right)}_{n}\\ {\mathrm{SiX}}_{3}+4H_{2}O\;\to \mathrm{Si}{\left(\mathrm{OH}\right)}_{4}\;\overset{-H_{2}O}{\to }{\;\mathrm{SiO}}_{2} \end{array} \end{array}$$

#### 3.1.2. Alkoxysilanes

Alkoxysilanes conform to H_n_Si(OR)_4−n_. The Si-O bond is much more reactive than the Si-C bond, especially when OMe or OEt is linked to the silicon atom. However, when the R in -SiOR has a large spatial site resistance, it becomes very inactive and also has good heat resistance when R is an aryl group.

H_n_Si(OR)_4−n_ can undergo hydrolysis reactions under certain conditions:$$\equiv \mathrm{SiOR}+H_{2}O\;\to \equiv \mathrm{Si}-\mathrm{HO}+\mathrm{ROH}$$

However, the hydrolytic activity of the Si-OR bond is much lower than that of Si-X. In addition, the structure of alkoxysilane and the hydrolysis conditions also greatly influence the hydrolysis reaction. The hydrolysis activity of Si(OR)_4_ decreases with an increase in the number of carbon atoms of R in OR. The commonly used catalysts for the hydrolysis reaction of Si-OC bonds are HCl, H_2_SO_4_, AcOH, Al_2_O_3_, ZnO, HNR_2_, and M(OH)_n_ (M is a metal). Chemical formula of organoalkoxysilane replaces the above —H with R (organic group), becoming R_n_Si(OR)_4−n_. Its properties are similar to alkoxysilane.

#### 3.1.3. Organohydrosilanes

Organohydrosilanes are broadly divided into two categories, those with only hydrocarbon and hydrogen atoms on the silicon atom, and those containing Si-Cl, Si-OR, or Si-OAc bonds. The second group is most commonly used because it can produce a variety of hydrogen-containing silicone oils, silicone rubbers, and silicone resins, as well as a range of carbon-functional alkyl groups through hydrosilation.

The addition of hydrosilanes containing Si-H bonds to unsaturated hydrocarbon groups and carbonyl groups is hydrosilation:$$\equiv \mathrm{Si}+{\mathrm{CH}}_{3}=\mathrm{CHR}\to \equiv {\mathrm{SiCH}}_{2}{\mathrm{CH}}_{2}R$$

#### 3.1.4. Organoyloxy Silanes

Organoxysilanes can be expressed as R_n_Si(OCOR′)_4−n_. The hydrolytic activity of organoxysilanes is intermediate between that of chlorosilanes and alkoxysilanes, and hydrolysis reactions can occur at room temperature without a catalyst:$$\equiv \mathrm{SiOAc}+H_{2}O\;\to \equiv \mathrm{SiOH}+\mathrm{AcOH}$$

The rate of hydrolysis of organoyloxy silanes increases with the number of acyloxy groups on the silicon atom.

#### 3.1.5. Haloalkylsilanes

In haloalkylsilanes, the reactivity of the C-X bond decreases in the following order: C-I > C-Br > C-Cl > C-F.

It can be found that the C-F bond is the least chemically active, but people synthesize fluoroalkyl silane because it can give excellent oil resistance, solvent resistance, hydrophobicity, stain resistance, etc. to polymerization.

#### 3.1.6. Silane Coupling Agents

The traditional silane coupling agent is a class of silane containing three hydrolyzable silicon functional groups; some of the silanes mentioned earlier also belong to the silane coupling agent. The silane coupling agent has three major functions. The first point is it can improve the bonding of the interface of the materials, playing the role of the adhesive and the coating adhesive. The second point is it can modify the surface of the material, giving the surface waterproof, anti-static, and anti-mold functions, as well as other functions. The third point is it can be used as a filler to increase the bonding between different chemical properties of materials, improving the mechanical properties of the product such as insulation, anti-aging, and hydrophobicity.

### 3.2. Polymerization of Silicones

The polymerization between silicones mainly relies on the dehydration condensation of silicone monomers. As mentioned before, some functional groups are easily hydrolyzed to produce silanols, which are dehydrated and condensed to obtain siloxanes, and it is the simplest and most effective way to synthesis polysiloxanes. The hydrolysis process can be expressed as follows:$$R_{3}\mathrm{SiX}+H_{2}O\;\to R_{3}\mathrm{SiOH}+\mathrm{HX}\;R_{2}{\mathrm{SiX}}_{2}+H_{2}O\;\to R_{2}\mathrm{Si}{\left(\mathrm{OH}\right)}_{2}+2\mathrm{HX}\;{\mathrm{RSiX}}_{3}+H_{2}O\;\to \mathrm{RSi}{\left(\mathrm{OH}\right)}_{3}+3\mathrm{HX}$$
where R is alkyl, aryl, alkenyl, etc., and X is halogen, alkoxy, and acyloxy. The hydrolysis rate of silico-functional silanes decreases in the order of halogen > acyloxy > alkoxy, and the hydrolysis rate of organohalogen silanes decreases in the order of I > Br > Cl > F. When trifunctional silanes are hydrolyzed, generally less than trisilanols are obtained, but rather condensed siloxanes are obtained.

#### 3.2.1. Hydrolytic Condensation of Difunctional Silicone Monomers

For example, the partial hydrolytic condensation of dialkyl dialkoxysilanes with small amounts of water produces high molecular weight polydialkyl siloxanes:

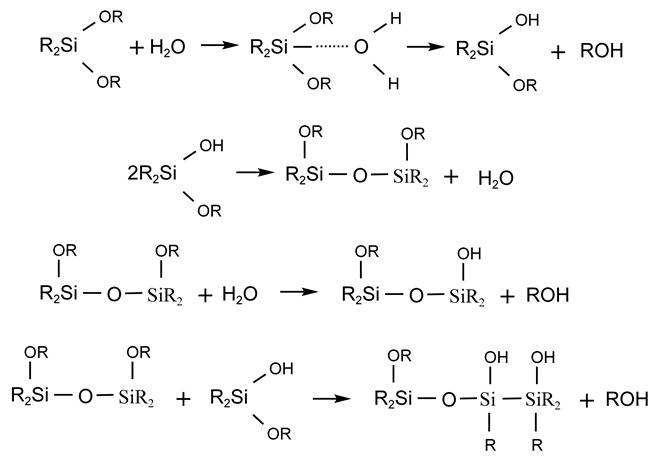


If the functional group organic monomer is hydrolyzed in excess water, a mixture of straight-chain and cyclic siloxanes is produced via dehydration polymerization. The proportion of the mixture is related to many factors: if an oxygen-containing reactive solvent is used, the proportion of cyclic siloxanes in the product increases; if straight-chain polysiloxanes are more easily produced under alkaline conditions, cyclic siloxanes are more easily produced under acidic conditions; and if the volume of substituents on the silicon atom is larger, cyclic siloxanes are more easily produced.

#### 3.2.2. Hydrolytic Condensation of Trifunctional Organosilicon Monomers

When trifunctional silicone monomers interact with small amounts of water, straight-chain siloxanes can be obtained, and the polymerization process is similar to that of dialkyl dialkoxysilanes.

Trifunctional organosilicon monomers undergo hydrolysis and condensation reaction in excess water, which can generate polymers with complex components, and cyclic compounds can be obtained in addition to cross-linked structures. For example, when methyltriethoxysilane is hydrolyzed and a condensation reaction occurs in excess water, the condensed cyclic compound can be separated:

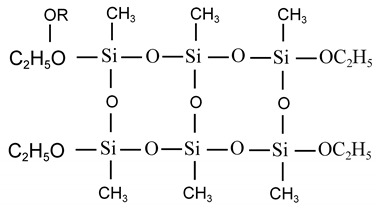


The higher degree of polymerization of the network structure requires further polymerization to complete the cross-linking, such as chain polysiloxane branched-chain condensation to complete the cross-linking, or cyclic polysiloxane ring-opening polymerization, or carbon tubular energy groups containing double bonds can be completed by hydrosilation to complete the cross-linking.

### 3.3. Surface Modification with Silicones

To our best knowledge, —C_n_H_2n+1_, —CH=CH_2_, —C_6_H_5_, —X, etc., are some representative hydrophobic groups, so it is possible to use the silane coupling agent, and the hydroxyl group on the surface to be treated is dehydrated and condensed; the hydrophilic group on the surface of the substrate hydroxyl disappears and is replaced by the hydrophobic group on top of the silane coupling agent, completing the hydrophobic treatment of the surface. If the surface substrate is a rough surface with a micro–nano structure, the grafting method can be used to complete the preparation of superhydrophobicity. This grafting method is used in many superhydrophobic preparation processes.

Zhao et al. reported a method to first prepare positively charged composite silica colloids using a simple multiple ultrafiltration method to form micro- and nanoscale roughness structures on metal substrates after modification of silica nanoparticles via electrochemical deposition [35]. The modification process is shown in Figure 4a. 3-Glycidoxypropyltrimethoxysilane (KH560) undergoes hydrolysis and then dehydrates and condenses with the hydroxyl groups on the silica surface. This way, the silica is grafted with KH560 around it. Fang et al. described a method for the synthesis of superhydrophobic nanosilica using glycidoxypropyltrimethoxysilane and dodecylamine as modifiers [36]. The authors added a mixture of fumed silica/toluene to a flask and sonicated it. After that, the mixture was treated with heat and KH560 was added dropwise via magnetic stirring. Subsequently, the mixture was heated and refluxed at 110 °C for 5 h. The hydroxyl groups on the surface of silica nanoparticles were grafted to epoxy groups. Subsequently, dodecylamine dissolved in toluene was added dropwise to the above suspension, and the entire process is shown in Figure 4b. Motornov et al. reported a method for preparing superhydrophobic nanoparticles by using a chemical bonding method to combine block copolymers on the surface of silica nanoparticles [37]. The authors used a “grafting” method to brush graft poly(styrene-b-2-vinylpyridine-b-ethylene oxide) and poly(styrene-b-4-vinylpyridine) block copolymers onto silica nanoparticles with two different diameters, as shown in Figure 4c. The grafting process was carried out in three steps. Kansara et al. reported a facile one-step dip-coating method for the preparation of recyclable superhydrophobic polypropylene membranes [38]. Superhydrophobic membranes with nanoscale surface roughness and a contact angle (water) greater than 150° were prepared by dipping polypropylene fabric films into a solution of silica nanoparticles grafted with silicone, and showed excellent oil–water separation efficiency and a relatively high flux in a gravity-driven separation system. As shown in Figure 4d, the authors used different trichlorosilanes grafted onto the hydrophilic silica during the experiments.

Peng et al. proposed a fast and favorable method for mass production to fabricate superhydrophobic and highly oleophobic surfaces with good mechanical durability [39]. These coatings are fabricated using dip and spray methods. In their experiments, the authors used ZnO nanoparticles and hydrophobic vapor-phase SiO_2_ nanoparticles to build micro-/nano-structures that enhance the roughness of superhydrophobic surfaces, while 1H,1H,2H,2H-Perfluorodecyldichloromethylsilane (FAS) was used to modify the surface of the nanoparticles to present low surface energy. Zhang et al. reported a method to modify the SiC surface from intrinsically hydrophilic to superhydrophobic nanoparticles using fluorinated organic substance FAS [40]. As shown in Figure 4f, the alcohol hydroxyl group of the hydrolyzed FAS and the hydroxyl group on the SiC surface undergo a dehydration reaction to form Si-O-Si. Xue et al. reported a method to modify fibers with mercaptosilanes [41] by grafting compounds containing hydrophobic groups onto the cotton fiber surface as shown in Figure 4g. The final prepared superhydrophobic textile possessed a contact angle of 159° and a rolling angle of 4°.

**Figure 4 polymers-15-00543-f004:**
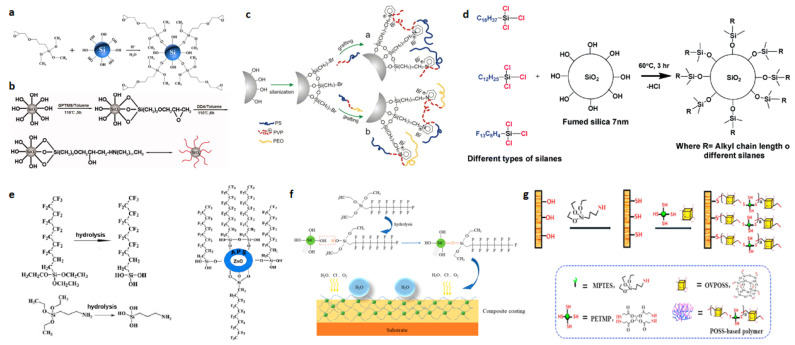
(**a**) Schematic illustration showing the synthesis route of 3-Glycidoxypropyltrimethoxysilane (KH560)-modified SiO_2._ Reprinted with permission from Ref. [35]. 2019, Appl. Surf. Sci. (**b**) Schematic formation of superhydrophobic nanosilica. Reprinted with permission from Ref. [36]. 2010, Polym. Compos. (**c**) Schematics of the grafting route to fabricate mixed-block copolymer brushes on the silica particle surface. Reprinted with permission from Ref. [37]. 2007, J. Colloid Interface Sci. (**d**) Reaction of silica with trichloro(alkyl)silane to form alkylsiloxane silica. Reprinted with permission from Ref. [38]. 2016, RSC Adv. (**e**) Hydrolysis of 1H,1H,2H,2H-Perfluorodecyldichloromethylsilane (FAS), hydrolysis of 3-aminopropyltriethoxysilane, and modification of ZnO nanoparticle surface with FAS. Reprinted with permission from Ref. [31]. 2016, J. *Nature*. (**f**) Experimental procedures of fabricating F–SiC and composite coating. Reprinted with permission from Ref. [40]. 2020, Coatings. (**g**) Schematic illustration of the fabrication of superhydrophobic fabrics. Reprinted with permission from Ref. [41]. 2019, Appl. Surf. Sci.

### 3.4. Silicones Build Micro- and Nano-Structures

#### 3.4.1. Stöber Method

Hydrophilic silica was generated using the Stöber method and then the silica was modified using other silane coupling agents while firmly bonding the silica powder to the substrate surface.

Zhao et al. reported a silica nanoparticle precursor dispersion prepared by the Stober process followed by surface modification to prepare a superhydrophobic surface [42]. The modification process utilizes the hydrolysis–condensation reaction of tetraethoxysilane (TEOS) and 1H,1H,2H,2H-perfluorodecyltriethoxysilane (HDFTES) to branch on fluorinated groups on top of the silica nanoparticles as shown in Figure 5a. Here, TEOS acts as both a reactant for silica and a modifier for hydrophilic silica, as well as a binder for the final surface. The surface after spraying possessed a contact angle of 150.8° and a rolling angle of 1.7°. Wang et al. reported a combined micro-arcoxidation (MAO) and sol-gel method using TEOS and methyltriethoxysilane (MTES) as precursors for the preparation of Mg-3.0Nd-0.2Zn-0.4Zr (wt, NZ30K) magnesium alloy on superhydrophobic silica films [43]. The authors used the Stöber method to generate silica particles, increase the roughness of the surface, and construct micro–nano structures, as shown in Figure 5b.

#### 3.4.2. PDMS

Polydimethylsiloxane (PDMS) possesses good elasticity and demolding properties, so PDMS is also often used to build micro–nano mechanisms.

Oh et al. reported a template method using PDMS to prepare superhydrophobic surfaces [44]. The authors formed regular micron-sized column arrays on silicon wafers using conventional lithography and inverted the cylindrical hole arrays of the silicon master into PDMS molds. The resulting films were separated from the molds without damage or deformation due to the elasticity, high chemical stability, and low interfacial energy of PDMS, as shown in Figure 5c. Wang et al. reported an inverted pyramidal structure of the polydimethylsiloxane sticker [45]. The authors used a template method to allow the elastic PDMS to form a material with a micro–nano structure, as in Figure 5d, and then used trichloro-(1H,1H,2 H,2H-perfluorooctyl) silane to bind a molecular layer to the PDMS surface, which facilitated the detachment of PDMS more easily.

#### 3.4.3. Polymerization of Silicones

According to the hydrolytic condensation of silicones, silicon–hydrogen reactions can then be cross-linked with each other to form rough structures.

Qu et al. reported a method for constructing multivacancy rough structures through organosilicon hinges [46]. A lightweight, mechanically flexible, and surface superhydrophobic organosilica aerogel was used to tune the cross-linked network via a simple environmental drying process. Polymethylhydrosiloxane with different hydrogen content was utilized to react with vinylmethyldimethoxysilane under the Pt catalyst, thus effectively tuning the cross-linking network in the organosilicon aerogel. The specific reaction is shown in Figure 5e. The prepared silicone aerogels not only have an excellent absorption capacity for various oils and solvents, but also have excellent continuous oil/water separation efficiency. Li et al. reported a simple method for the preparation of robust, compressible, bendable, and stretchable silicone sponges [47]. The silicone sponges were subjected to solvent-controlled hydrolysis of silane in the presence of water condensation in the presence of water, followed by drying at 60 °C under ambient pressure. The mechanical properties of silicone sponges are closely related to the network structure of their composition, which can be controlled by co-solvents with different polarities (e.g., alcohols, alkanes, and aromatics).

## 4. Silicone-Based Water-Repellent Surface Preparation

As already mentioned in the review, silicones can be cross-linked to build micro–nano structures via dehydration condensation, and short-chain self-assembled monolayers can also be formed to reduce the surface energy without changing the surface structure morphology. Therefore, silicone-based water repellent surfaces can also be prepared using many methods, such as electrostatic spinning [48], spraying [49], templating [50], particle coating [51], chemical deposition [52], etc.

### 4.1. Electrostatic Spinning

Electrostatic spinning is a simple but versatile method for producing continuous fibers with diameters ranging from nanometer to submicron scales. The principle is to use a high DC electric field or high electric force to overcome the surface tension on the surface of the polymer solution and produce very thin jets. Therefore, electrostatic spinning is very suitable for preparing superhydrophobic materials, and silicone materials also play a key role in the preparation of many electrostatic spinning processes.

Organic silicon materials can be blended with their organic polymers as electrostatic spinning feedstock to prepare superhydrophobic materials. Song et al. reported an electrostatic spinning method to prepare superhydrophobic materials with polyvinylidene fluoride (PVDF) [48]. The authors added TEOS to the electrostatic spinning solution of PVDF, and then applied a voltage to complete the electrostatic spinning precursor solution. When the electrospun nanofibers were annealed at 100 °C under room temperature, the condensation reaction was accelerated to form a cross-linked silica network and PVDF was embedded in the TEOS network through chemical bonding chains, as shown in Figure 6a. The contact angle of the final prepared superhydrophobic material reached 155.5°. Chen et al. reported electrostatic spinning using fluorinated silane-functionalized polyvinylidene fluoride [53]. The authors used perfluorooctyltriethoxysilane to attach poly(chlorohydrogen triethoxysilane) to a PVDF polymer under acidic conditions to make it hydrophobic, as shown in Figure 6f; coupled with the high roughness inherent on the polymer, it was arbitrarily able to form superhydrophobic PVDF films with contact angles greater than 150°. Sampathkumar et al. reported a simple one-step electrostatic spinning process [54]. TEOS was mixed with water and ethanol in a molar ratio of 1:3:8 and a drop of HCl was added as a catalyst. The mixture was stirred at 60 °C for 1 h and allowed to stand for different durations (1, 2, 3, and 5 h) to obtain a precipitated solution. After the completion of sedimentation, the solution was mixed with a 7% solution of PVA. The final superhydrophobic surface was prepared, as shown in Figure 6b.

Organosilicon can also modify some of the raw materials hydrophobically and then perform electrostatic spinning experiments in configuring the reaction solution. Mookgo et al. reported a method to prepare superhydrophobic-superoleophilic nanofiber membranes via electrostatic spinning [55]. Perfluorooctyltriethoxysilane was deposited on carbon nanotubes (CNTs). Then, the treated CNTs were added to a PVDF polymer solution for electrostatic spinning to finally prepare superhydrophobic fiber membranes, the flow of which is shown in Figure 6c. This nanofiber membrane has good oil–water separation capability. Lebea et al. reported a method to prepare high flux nanofiber membranes [56]. The surface of silica was modified with three different silane reagents octadecyltrimethoxysilane, N-octadecyltrichlorosilane, and chloro-dimethyloctadecyl silane, and finally the silica was encapsulated on PVDF nanofiber membranes using electrostatic spinning technique, and the contact angles of the obtained nanofiber membranes were significantly enhanced (156.4 ± 2.4°, respectively; 162.6 ± 1.8° and 151.7 ± 2.1°, respectively).

**Figure 6 polymers-15-00543-f006:**
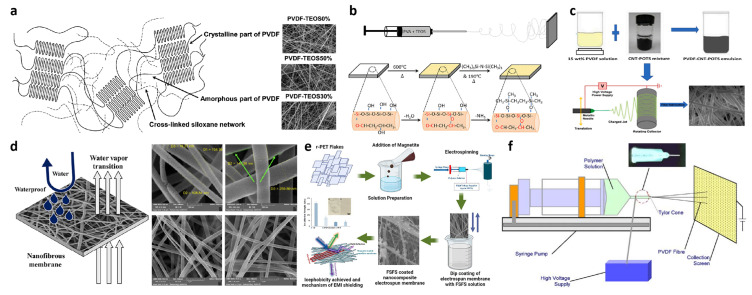
(**a**) Schematic of the cross-linking of TEOS among polyvinylidene fluoride (PVDF) chains and the SEM of PVDF-TEOS0%, PVDF-TEOS30%, and PVDF-TEOS50%. Reprinted with permission from Ref. [48]. 2010, ACS Appl. Mater. Interfaces. (**b**) Schematic representation of surface-modified nanofibrous coating. Reprinted with permission from Ref. [54]. 2021, Appl. Polym. Sci. (**c**) Synthesis process of nanofiber membrane. Reprinted with permission from Ref. [55]. 2022, Environ. Chem. Eng. (**d**) Scheme of a fibrous membrane with water vapor transition and the SEM of fibrous membrane. Reprinted with permission from Ref. [57]. 2019, Appl. Polym. Sci. (**e**) Magnetite/r-PET nanocomposite electrospun film process. Reprinted with permission from Ref. [58]. 2022, Polymer (**f**) Scheme of setup of electrospinning system. Reprinted with permission from Ref. [53]. 2009, Appl. Surf. Sci.

Silicones can be silanized on their rough surfaces after the completion of electrostatic spinning to complete the preparation of superhydrophobic fiber membranes. Faride et al. reported a combination of electrostatic spinning techniques and the dichlorodimethylsilane deposition method to prepare superhydrophobic nanofiber membranes [57]. The electrostatic spun nanofiber membranes formed from PVDF were modified with dichlorodimethylsilane (DMDCS) reagent to successfully prepare superhydrophobic and super lipophilic functionalized electrostatic spun PVDF membranes, as shown in Figure 6d. DMDCS in the membranes improved the antifouling properties, water resistance, and permeability, and accelerated oil–water separation. Mahmut et al. reported a method to prepare nanocomposite electrostatic spun membranes using superhydrophobicity and EMI shielding [58]. Magnetite nanocomposite nanofiber membranes were prepared using an electrostatic spinning process, after which the electrostatic spun membranes were immersed in a hexane solution consisting of 5 mg/mL modified nanoparticles for 5 s and then heated at 120 °C for 1 h, as shown in Figure 6e. Among the modified nanoparticles, SiO_2_ was modified by 1H,1H,2H,2H-Perfluorooctyltriethoxysilane as a silane coupling agent.

### 4.2. Spray Method

The spray method is a working method in which the coating is pressed or sucked out of the container by an external force and forms a mist to adhere to the object’s surface. In the superhydrophobic preparation process, silicones can either bond the micro–nano particles in the spray solution together or hydrophobically modify the surface.

Hydrophobically treated SiO_2_ nanoparticles are commonly used in spray coating methods. Ge et al. reported a one-step spraying method for preparing superhydrophobic colored films [59]. Arrays of monodisperse fluorinated silica nanoparticles were prepared using a one-step spraying method, as shown in Figure 7a. The silica NPs were prepared using the Stöber reaction and hydrophobized using fluoroalkylsilanes ((heptadecafluoro-1,1,2,2-tetrahydrodecyl)dimethylchlorosilane), resulting in a superhydrophobic coating with a rolling angle of less than 2°. Because of the environmental problems of fluorinated silanes, many works also choose other silanes for substitution, and Subramanian et al. reported a simple and fast spraying method [60]. The SiO_2_ nanoparticles were modified using hexamethyldisilazane, and a silane coupling agent (3-aminopropyl)-triethoxysilane was added to the spray solution to act as a binder, as shown in Figure 7b. The coatings obtained via spraying exhibited a water contact angle > 150° and a sliding angle < 10°, which remained unchanged even after 30 wear cycles. Mohseni et al. reported a two-step spraying method [61]. Spray 1 contained colloidal-stabilized silica nanoparticles synthesized via the sol-gel method and Spray 2 contained low-surface-energy alkyl silane (hexadecyltrimethoxysilane) and PDMS-APS, which can be sprayed in two steps to prepare wear-resistant superhydrophobic coatings on textiles, as shown in Figure 7c.

In addition to the use of silica nanopowders, many micro–nano particles can be used in spraying methods. Feng et al. reported a spraying method using a halloysite nanotube [62]. In this method, n-cetyltriethoxysilane- and tetraethoxysilane-modified halloysite nanotubes were used, using the dehydration condensation of both culminates in the formation of a superhydrophobic surface as shown in Figure 7d. Peng et al. reported an anti-icing superhydrophobic layer that could be used on asphalt surfaces [63]. The surface of magnesium–aluminum-layered double hydroxides was modified using γ-methacryloxypropyltrimethoxysilane (KH550), in which tetraethyl orthosilicate was used as a cross-linking agent and dibutyltin dilaurate was used as a catalyst for cross-linking, as shown in Figure 7e. The final superhydrophobic coating was prepared on asphalt, which significantly delayed the freezing time of the surface water and also possessed good durability. Yu et al. reported a spraying method using alumina particles [64]. Different superhydrophobic surfaces were prepared using 1H,1H,2H,2H-perfluorodecyltrimethoxysilane-, MTES-, and diethoxydimethylsilane (DMDES)-modified alumina nanoparticles, respectively, to investigate the differences between their properties.

In the spraying method, compared with some common superhydrophobic reagent preparations, the advantages of superhydrophobic coating using silicone raw materials are as follows: the dehydration condensation reaction of silicone is well controlled, the reaction solution is partially condensed to form a prepolymer, and the structure of the prepolymer is stable and can be kept in the container for a long time, avoiding the trouble of ready-to-use first preparation. It also reacts rapidly with atmospheric water when spraying and forms a super hydrophobic coating.

### 4.3. Template Method

The stencil method is a commonly used method for preparing superhydrophobic coated films, which is a monolithic coverage surface technology. The template method uses a solid with a rough structure as a template, and the hydrophobic material is formed and demolded on the rough solid surface via extrusion or coating followed by light curing on a specific template to produce a superhydrophobic film. The preparation of superhydrophobic coatings using the template method has the advantages of simple operation, good reproducibility, and controllable nanowire diameter ratio. In the preparation of superhydrophobic water using the template method, PDMS demonstrates its excellent characteristics because PDMS is flexible and can be easily peeled off during the demolding process while the processed PDMS keeps the mold intact and undamaged, and PDMS is flexible and contacts the relatively rough surface very closely after processing. So, almost superhydrophobic materials on silicone substrates use PDMS when using the template method.

PDMS also has different roles in the preparation process, and a large part of it is PDMS as a product as a result of forming a superhydrophobic surface with a PDMS substrate. Generally, templates with micro–nano structures are prepared via photolithography, laser etching, etc., and PDMS is injected and cured. Sun et al. reported a hydrophobic anti-icing surface with a multifunctional surface prepared using the template method [65]. A mixture of PDMS and curing agent mixed in proportion was poured onto the prepared square CNW template and the internal gas was removed in a vacuum chamber. To increase the surface coarseness, these CNW templates were laser etched. Afterward, the mixture was placed in an oven for 2 h to cure and was demolded to complete the preparation, as shown in Figure 8a. The final superhydrophobic surfaces prepared by the authors achieved a contact angle of 155.4 ± 0.5° and excellent surfaces in terms of anti-icing. Han et al. reported a combination of UV curing and the template method to prepare superhydrophobic surfaces with a good anti-fogging effect [66]. A negative photoresist SU-8 was spin-coated on a clean slide and a template containing a circular array was photolithographed. A mixture of PDMS prepolymer and curing agent in a 10:1 mixture (by weight) was poured on top of the template and then degassed using an air pump; after curing at 8 °C for 1 h, the PDMS samples finished curing and could be peeled off from the mold, as shown in Figure 8b. Finally, the authors demonstrated that the prepared superhydrophobic layers had a good anti-fogging effect. Pan et al. reported a scheme to prepare fully sparse PDMS surfaces using a template method [67]. Wet chemical etching was first used to fabricate a series of Si nanowire molds with different roughness. Silica nanoparticles were combined on the mold surface to enhance the roughness of the surface microstructure. After PDMS was injected, cured, and demolded, the surface was modified with low-surface-energy trichloro (1H,1H,2H,2H-perfluorooctyl) silane. The final PDMS micropillar structures exhibited excellent hydrophobic properties, not only in various liquids with static contact angle (≥150°) and sliding angle (≤6°).

There is also much work to be conducted because the final product requires other property materials, such as memorable materials and some materials that are sensitive to external stimuli but do not have good demoldability such as PDMS, and often PDMS is used as an intermediate template to prepare functional superhydrophobic surfaces. Lin et al. reported a scheme to prepare superhydrophobic surfaces with magnetically induced low-adhesion orientation using the template method [50]. The PDMS negative structure array was first replicated via soft molding to the Si micropillar array as a template in the second step, and then the NdFeB/PDMS composite was poured onto the PDMS mold and the final PDMS material after curing and demolding, i.e., that has a micro–nano structure and contains magnetic particles, as shown in Figure 8d. The final superhydrophobic surface can be modified by applying a magnetic field to change the surface hydrophobicity. Chandra et al. reported a method to control the roughness of superhydrophobic surfaces via a template method [68]. Various-shaped micropillars including conical, circular, and square cylinders were first replicated from the corresponding silicon master molds using PDMS. The monomers were mixed with photoinitiators to first partially cure them to obtain viscous molding precursors. The precursors were injected on top of the various PDMS models obtained above and fully cured using UV light to finally obtain hydrogel micropillar arrays, as seen in Figure 8e. Superhydrophobic surfaces responsive to the environment were obtained, which could be applied to sensors, responsive surfaces, and biological studies. Pan et al. reported a scheme for preparing smart superhydrophobic surfaces with self-healing surface chemistry using a template method [69]. PDMS microcolumn arrays were first prepared as templates for the second step; the bisphenol epoxy resin E-51 and its cross-linking agents m-xylylenediamine and n-octylamine were mixed and blended. The mixture was poured on top of the PDMS templates and pumped using a vacuum pump until no bubbles appeared on the surface. Then, it was cured at 60 °C for 2 h and at 100 °C for another 1 h. After completing the curing, it was demolded, as in Figure 8f.

The flexibility and good release properties of PDMS are unsurpassed by most superhydrophobic materials. In preparing superhydrophobic materials using the template method, PDMS needs to be included regardless of whether the final superhydrophobic material is silicone.

### 4.4. Particle Filled Method

The particle filled method is a direct combination of silicone and various nano-powders to form a superhydrophobic surface with a particle-based surface. The superhydrophobic surface of the silicone substrate is mainly formed by silica prepared using the Stöber method, or silicone acts as a binder and modifier in other nano-powders to form a superhydrophobic surface. This method has the advantage of simple and convenient operation, but because the powder is the main body, it tends to lead to poor surface durability and wear resistance.

Yu et al. described a new method for creating superhydrophobic thin films [70] in which polyethylene was synthesized on self-assembled colloidal silica nanoparticles combined with a single-site Cr-oxide catalyst. Due to the combined effect of the hydrophobicity of the synthesized polyethylene and the microstructure formed by the colloidal silica nanoparticles, the obtained films showed a static water contact angle of 162°, as shown in Figure 9a. The surface was polymerized silica. Jradi et al. reported a method to prepare superhydrophobic by stacking SiO_2_ powder [71]. The resulting silica powder was deposited using the Stöber method attached to an aminosilylated surface, in which the surface was amidated using KH550. Finally, the surface silica powder was modified using a n-octadecyltrichlorosilane solution, which is the process of linking the branched chains mentioned earlier. This exposes the surface as “—(CH_2_)_18_”, and together with the rough structure the final result is a superhydrophobic surface with a contact angle greater than 150°, which is prepared as shown in Figure 9b.

Lu et al. reported a method to modulate the transition from hydrophilic to superhydrophobic surfaces [72]. Highly transparent superhydrophobic surfaces were prepared using a commercial silicone sealant and functionalized silica nanoparticles. The mixture was obtained by mixing and stirring SiO_2_ nanoparticles and silicone sealant followed by sonication. This mixture can be easily prepared as a coating on most substrate materials, including iron alloys, wood, concrete, glass, and tiles, as shown in Figure 9c. The main component of the silicone adhesive here is α,ω-dihydroxydimethylpolysiloxane. During the preparation process, the silicone acts as a bonding powder as well as a modified surface, and the contact angle of the final product can reach 169°. Allahdini et al. reported a method to prepare transparent fluorine-free superhydrophobic coatings [73]. Using alkoxysilanes as binders, fumed silica nanoparticles and MTES were combined to form a robust transparent superhydrophobic coating. MTES can act as a coupling agent between the nanoparticles and the binder to improve the stability of the layered structure, as shown in Figure 9d. The CA of the developed superhydrophobic coating is 163° ± 2.1°. Even at very low surface slopes, water droplets placed on the superhydrophobic coating roll off the surface quickly. Li et al. reported a “glue + powder” approach to prepare superhydrophobic surfaces [24]. Using hexane to dilute commercial PDMS to form a thin layer of glue at the base, more than 17 types of particles were sprinkled directly onto the uncured adhesive surface, and it was found that any kind of particle could eventually be prepared into a superhydrophobic surface. The specific principle is shown in Figure 9e. The method is simple and fast and facilitates large-scale production. Xuan et al. reported the preparation of a corrosion-resistant superhydrophobic surface [74]. The nanoparticles were first modified using silicone, and then the modified nanoparticles were fed into N-2-aminoethyl-3-aminopropyltrimethoxysilane (1 w%), dodecylbenzenesulfonate (0.5 w%), and ammonia (2.5 w%). After stirring the mixture for 2 h, the mixture was directly coated on the Cu-based surface and cured to complete the superhydrophobic preparation. Here, the silicone plays the same role of bonding and modification.

In this section, silicones were shown to serve different functions compared to other superhydrophobic reagents. Silicones act as binders between nanoparticles and substrates, and also as modifiers between nanoparticles and the final coating surface. Meanwhile, silicones can react to generate silica powders that provide micro- and nano-structures for superhydrophobic surfaces.

### 4.5. Chemical Deposition Method

Chemical deposition methods are widely used to prepare superhydrophobic surfaces because they can directly and effectively construct suitable surface roughness and reduce surface energy. The preparation of superhydrophobic surfaces on silicone substrates via chemical deposition is usually accompanied by chemical reactions, such as the cross-linking of silicone polymerization and grafting rough surfaces. They can be divided into vapor deposition, solution deposition, and electrochemical deposition.

Zhang et al. reported a superhydrophobic preparation method using trimethylchlorosilane (TMCS) for chemical deposition [52]. Tetraethyl orthosilicate was used to prepare hydrophilic silica nanoparticles, which were assembled onto glass sheets using the spin-coating method to complete the construction of micro–nano structures. Then, solution deposition was performed using 4 *v*/*v*% TMCS hexane solution to change the surface from a hydrophilic surface to a superhydrophobic surface, as shown in Figure 10a. The prepared superhydrophobic surface had good hydrophobicity, superhydrophobicity, and transparency. Huang et al. reported a method using vapor phase deposition that allowed for the green preparation of composite superhydrophobic coatings [75]. The preparation of the composite coating was accomplished by dissolving cellulose nanocrystal particles into an aqueous solution of PVA. The coating was prepared on top of the wood using the vapor deposition method to reduce the surface layer as shown. One of the modifiers was 1H,1H,2H,2H-perfluorooctyltrichlorosilane, as shown in Figure 10b, and the final superhydrophobic coating possessed a contact angle of up to 158.2°. Wang et al. reported a scheme to improve the corrosion resistance of magnesium alloys using electrochemical deposition [76]. As shown in Figure 10c, the electrolyte for electrochemical deposition was an ethanol electrolyte of cetyltrimethoxysilane and tetraethyl orthosilicate, and the diatomaceous earth powder was successfully modified after electrochemical deposition. Wang et al. reported a self-healing oil–water separation superhydrophobic coating [77]. As shown in Figure 10d, firstly, octadecyltrimethoxysilane was dispersed into pure water using ultrasonication; then, dopamine and ammonia were added and were ultrasonicated for 30 min to complete the solution treatment, after which the textile was prevented from solution deposition with the solution to complete the superhydrophobic treatment. Bai et al. reported a method to achieve efficient water collection on a surface by combining wettability and shape gradients [78]. As shown in Figure 10e, the superhydrophobic surface was prepared by first depositing TiO_2_ particles on top of a glass plate and then chemically depositing the surface using FAS. According to the photocatalytic decomposition nature of TiO_2_, the patterning of the superhydrophobicity can be accomplished using area irradiation with UV light. Li et al. reported a superhydrophobic coating with a cactus structure [79]. The cactus-like coarse superstructure was engraved directly on the aluminum plate using a laser, and then the surface was modified via vapor deposition of 1H,1H,2H,2H-perfluorodecyl trichlorosilane. The surface of the prepared coating exhibited excellent hydrophobicity with a contact angle of 161.2°.

This section aims to reflect the great advantages of silicones in the preparation of superhydrophobic materials; silicones can be combined with most materials to achieve modifications and not be limited to low-surface-energy substances, thus vastly expanding the range of material options when designing and preparing superhydrophobic surfaces.

Furthermore, in some studies, the superhydrophobicity of silica preparations was combined with engineering customization, such as the controllability of superhydrophobic properties by adjusting the parameters of laser [80] and lithography [81]. Carrascosa et al. varied the superhydrophobicity by changing the power of the laser [82]. The researchers’ hydrophobized marble slabbed with PDMS and other silica columns affected the roughness of laser engraving by varying the laser power. The power was 0%, 60%, 70%, 80%, 90%, and 100%, and the final roughness was 6.5, 6.6, 15.5, 18.5, 25.3, and 20.8 microns, respectively. The contact angle and roughness of the treated products were positively correlated, with 80% and 90% of the contact angles exceeding 150, achieving superhydrophobic performance. Qi et al. reported a method for laser engraving on silicone surfaces [83]. The degree of surface wetting was controlled programmatically by varying the grayscale. The corresponding contact angles for materials with grayscale values of 10%, 14%, and 18% were 152.5°, 137.2°, and 125.7°, respectively. Zhu et al. introduced photolithography to control the surface wettability of silica gel by varying the micro-pillar spacing and thus the surface wettability [84]. The side length of the micropillars remained at 10 µm, and the distance between the micropillars was increased continuously (b = 5, 10, 15, 20, 25), which corresponded to a gradual increase in hydrophobicity (contact angle = 147.6, 151.9, 153.4, 153.3, 161.1). Combining these engineered customizations enables the enhanced controllability of silicone superhydrophobic preparation and a more comprehensive range of applications.

## 5. Silicone-Based Superhydrophobic Applications

Silicone-based superhydrophobic materials also have a large number of applications, the most common being self-cleaning [85], oil–water separation [86], anti-icing [87], and corrosion protection [88].

### 5.1. Self-Cleaning

Self-cleaning is the most basic application of superhydrophobic surfaces. After being contaminated, the superhydrophobic surfaces can take the contaminants away by themselves when rainwater rolls over the surface to achieve self-cleaning. The principle is that on a rough, low-energy surface, if the surface is contaminated with dust particles, the interface area between the contaminated particles and the surface is relatively small, resulting in reduced adhesion. Figure 11a shows a schematic representation of the self-cleaning effect [89]. Water droplets from rainfall can absorb contaminated particles and carry them away as the droplets roll off the surface, achieving a self-cleaning effect. Simulated pollution experiments on various natural plant leaves (naturally hydrophobic surfaces) show that pollutant particles on hydrophobic superhydrophobic leaves are almost completely removed by water droplets, while a considerable number of particles remain on non-superhydrophobic leaves.

Pawar et al. reported a method that used surface protection and self-cleaning [85]. Different concentrations of modified silica particles were used along with polystyrene to develop superhydrophobic coatings with self-cleaning and the silica particles were first modified using methyltrichlorosilane. Then, the modified silica particles and polystyrene were mixed and spin-coated on top of the substrate. The resulting superhydrophobic layer has an excellent self-cleaning function, as shown in Figure 11b. When there is an inclined angle, the water droplets easily carry away the contaminants in the trajectory, and on the horizontal superhydrophobic surface, the water droplets automatically absorb the surrounding contaminants. Gong et al. reported a method to prepare a highly durable superhydrophobic with good self-cleaning capability [90]. Hydrophilic SiO_2_ nanoparticles and hydrophobic silicon carbide nanoparticles were mixed in a certain ratio and blended with PDMS, and curing was completed after heating at 80 °C for 1 h to obtain a multistage micro- and nano-structured surface, where PDMS was used as both a low-surface-energy modifier and a binder. The authors chose black carbon powder as a model dust and placed it on the film surface in order to demonstrate the self-cleaning properties of the prepared superhydrophobic films. The superhydrophobic film was tilted at a certain angle in a Petri dish. Then, water drops were placed on the superhydrophobic film, and the water drops could absorb and carry away the model powder, as shown in Figure 11c. Wang et al. reported a fluorine-free superhydrophobic coating with exotic self-cleaning and bouncing properties [91]. Physical deposition was used to fabricate the Al_2_O_3_/RTV silicone rubber superhydrophobic surface. A layer of Al_2_O_3_ was uniformly dispersed on the surface of the incompletely cured RTV silicone rubber, and the excess nano-powder was blown away using an inert gas, and ultrasonic cleaning was performed after the completion of curing. After drying, the superhydrophobic surface was obtained, and the superhydrophobic surface had excellent self-cleaning properties. As shown in Figure 11d, the liquid droplets easily carry away the graphite powder from the coated surface. Peng et al. demonstrated a simple method to prepare durable superhydrophobic surfaces using two types of nanoparticles [39]. Epoxy resin was used as an adhesive material to improve the wear resistance of the surface. ZnO nanoparticles and SiO_2_ nanoparticles were used to produce high surface roughness. The nanoparticles were treated with FAS and the prepared surfaces exhibited excellent superhydrophobicity. As shown in Figure 11e, the solid contaminants (fine sand) on the superhydrophobic surface were carried away when the liquid rolled off the coated surface.

### 5.2. Corrosion Resistant

Superhydrophobicity has good corrosion resistance; on the one hand, the air film in the superhydrophobic micro–nano structure will directly isolate the metal from the external corrosive solution. On the other hand, when exposed to air, metals tend to form electrolyte films easily on the surface, and the formation of such electrolyte films can lead to a sudden increase in the corrosion rate as they are carriers of electrons, oxygen, and carbon dioxide [92]. On superhydrophobic surfaces, the droplets produced after condensation are separated from each other by gas films in the surface roughness and electrons cannot move freely. As a result, electrochemical reactions are hindered, leading to superhydrophobic surfaces with good corrosion resistance [93].

Wang et al. proposed a simple and low-cost method for preparing superhydrophobic surfaces on aluminum substrates using a two-step spraying method [94]. In the first step, the substrate was sprayed with a layer of hydrocarbon resin binder, and in the second step, dichlorodimethylsilane-modified hydrophobic silica nanoparticles were sprayed on the binder. To complete the preparation of superhydrophobic, silica mainly plays a hydrophobic role in the experiment. The prepared superhydrophobic surface possesses good corrosion resistance. Figure 12a shows the Tafel polarization curves of superhydrophobic surfaces and untreated smooth surfaces in 3.5 wt% NaCl aqueous solution; Tafel polarization curves are commonly used for corrosion resistance experiments, in which the more hit the corrosion voltage, the better the corrosion resistance line. Jin et al. reported a sol-gel method using TEOS and vinyltriethoxysilane as co-precursors [95], which can be used in a room-temperature environment to prepare onto an aluminum substrate. This superhydrophobic surface possesses good corrosion resistance, as shown in Figure 12b, which shows the Tafel polarization curves at different times after treatment, indicating that this corrosion-resistant surface decreases with time, but is always higher than the blank group. Jiang et al. reported the development of a smart protection system with superhydrophobicity and active self-healing capability on magnesium alloys through an integrated MAO and electro-assisted sol-gel deposition method [96]. The porous MAO precoat acts as a “shield” and a “reservoir”, respectively, to obtain enhanced corrosion resistance and sufficient inhibitor loading. The top silica skeleton is responsible for the superhydrophobicity of the surface, which is used to block electrolyte intrusion and avoid the premature leaching of inhibitors. A schematic diagram of the corrosion protection mechanism of the bifunctional composite coating is shown in Figure 12c. Song et al. reported a fluorine-free superhydrophobic concrete coating for corrosion protection [97]. Silicate cement and DC-30, a commercial water-based protectant whose main components are silane and siloxane, were mixed and stirred for 10 min. The prepared solution was then poured on top of the concrete to complete the coating preparation. Electrochemical corrosion experiments were performed by applying 26 V to the O-concrete coating and S-concrete in seawater for 20 min. As shown in Figure 12d, the O-concrete corroded severely as the electrochemical corrosion time increased; however, the electrolyte remained transparent and translucent throughout the electrochemical corrosion process due to the presence of a superhydrophobic corrosion-resistant coating on S-concrete. Guo et al. reported a method to prepare superhydrophobic silica coatings with excellent corrosion resistance using a sol-gel process [98]. The silica was modified with FAS, which eventually possessed good corrosion resistance, as shown in Figure 12e which is the Nyquist diagram of tinplate-epoxy/F-SiO_2_, where a larger radius of curvature of the curve indicates a better corrosion resistance line of the surface, from which it can be seen that the prepared superhydrophobic surface is possessing good corrosion resistance. Liu et al. reported a superhydrophobic coating with excellent mechanical durability and chemical stability [99]. The composite particles were modified with 1H,1H,2H,2H-perfluorodecyltrimethoxysilane and dispersed in TPU to obtain the TPU/CNTs@SiO_2_ composite coating. To accurately analyze the impedance, the EIS results were fitted using the equivalent circuit model shown in Figure 12f. It can be found that the capacitive ring diameter of TPU/CNTs@SiO_2_ is the largest, indicating that the composite superhydrophobic material is by far the best for corrosion resistance.

### 5.3. Oil-Water Separation

Oil-water separation is generally gravity-driven, based on the difference in surface wettability for water and oily liquids. Most superhydrophobic surfaces are hydrophobic and lipophilic, allowing oily liquids to pass and blocking the passage of water.

Luo et al. demonstrated the simple fabrication of a microcapsule with durable superhydrophobic and superoleophilic properties [87], which allows the effective separation of small oil droplets in mixtures or emulsions. The surface microcapsules were modified by treatment with (3-aminopropyl)triethoxysilane and TEOS to improve the hydrophobic properties of the microcapsules. These microcapsules showed good oil–water separation capability with high separation efficiency after filtration and durable UV lamp treatment. The separation process is shown schematically in Figure 13a. Liu et al. reported a special method to prepare superhydrophobic and superoleophilic 3D graphene-based materials (GF) [100]. The GF was prepared using a simple filter foaming method and, subsequently, the surface was modified with a silane coupling agent to obtain superhydrophobicity and superoleophilicity; the modifier used was a fluorinated 1H,1H,2H,2H-Perfluorodecanethiol. It can be seen from Figure 13b that the oil–water mixture would be separated by the prepared superhydrophobic film under the effect of gravity. The above shows that the prepared graphene foam has great potential for oil–water separation.

Xue et al. reported a superhydrophobic and superoleophilic textile for oil–water separation [101]. The superhydrophobic and superoleophilic textiles were fabricated by coating PET fiber membranes with hydrophobic silica sols. The results showed that the textiles with superlipophilic and superoleophobic properties possessed good oil–water separation, as shown in Figure 13c. The authors mixed dyed water and gasoline into a tube, and a large amount of oil drained vertically through the textile into the beaker due to gravity. However, the water rolled over the textile and flowed horizontally through the tube into the water collector. Subramanian et al. reported a fluorine-free superhydrophobic spray coating for oil–water separation applications [60]. HMDS-modified SiO_2_ nanoparticles were used with acrylic resin, linked by a silane coupling agent to form superhydrophobic nanocomposites. The composites showed high hydrophobicity and oil–water separation capability. Figure 13d shows a typical gravity-driven separation process, where toluene li easily penetrates the filter paper while water is repelled by this surface, thus completing the oil–water separation. Yang et al. reported on superhydrophobicity with durable and efficient oil–water separation [102]. Superhydrophobic fiber fabric was prepared using simple pyrolysis, an in situ hydrothermal reaction, and the hydrophobization method. The authors fabricated a filtration device with this superhydrophobic fabric as an intermediate filter layer to verify the oil–water separation function. The prepared equal volume of the oil–water mixture (hexane, toluene, hexadecane, diesel, and dichloromethane) was then poured on the upper glass tube for oil–water separation, as shown in Figure 13e. It can be well separated from hexadecane/water and dichloromethane/water. Mi et al. reported a superhydrophobic silica sponge for oil–water separation [103]. Cobalt (Co) nanoparticles were deposited on top of silica fibers via controlled precipitation to provide magnetic properties to the prepared material. PDMS was coated on the surface of modified silica sponges (MSSs) to reduce the surface energy of MSSs. After PDMS curing, the MSSs were superhydrophobic/superoleophilic and could be directly used for remote oil absorption and oil–water separation. As shown in Figure 13f, the absorption ability of MSSs for heavy oil (chloroform) and light oil (hexane) was explored, which proved that they possess good oil–water separation ability.

### 5.4. Anti-Icing

An ideal superhydrophobic surface should have a long delayed icing time [104] and a small ice adhesion [105] so that the ice formed on the surface can be removed by its own gravity or natural wind. Therefore, superhydrophobic surfaces possess good anti-icing capability.

Huang et al. reported a new method [106] that enables the creation of low-surface-energy polymer coatings from ordered nanoscale conical arrays using a CVD process. The polymer coatings are then plasma treated in a vacuum chamber, where the plasma treatment activates surface hydroxyl groups to facilitate grafting with silicones. The surface modification is completed by using trichloroethylene-based silane vapor to graft vinyl onto the polymer surface. As shown in Figure 14a, the icing delay times of NC-2, NC-3, and NC-4 were greatly increased compared to NC-1. Because their micropillar densities increase sequentially, the final hydrophobic properties formed also increase sequentially, proving that the better the superhydrophobic properties are, the more they can delay icing. Allahdini et al. proposed a method [73] that allows the preparation of transparent superhydrophobic coatings by combining silicone resins containing methoxy functional groups, fumed silica nanoparticles, and MTES. This method can be applied to various substrates such as glass and metals. The final superhydrophobic coatings have good ice resistance. As can be seen in Figure 14b, the superhydrophobic layer icing time is 483 s, which is much longer than the control blank group of 15 s. Qi et al. proposed a method to prepare superhydrophobic surfaces with desirable anti-icing properties [107]. The superhydrophobic surfaces were successfully obtained using a combination of specific micro-/nano-structures and chemical modifications. As seen in Figure 14c, the best superhydrophobic surface prepared using this method was able to possess an icing delay of 98 ± 2.3 min, indicating that the material has ideal anti-icing properties and superhydrophobicity. Chao et al. reported a superhydrophobic coating on asphalt [63]. First, the surface of magnesium–aluminum-layered double hydroxide particles was modified with KH550. Then, the modified magnesium–aluminum-layered double hydroxides were mixed with RTV compounds and sprayed on the asphalt substrate. As shown in Figure 14d, the silicone-treated asphalt surface possessed a good anti-icing effect. Zeng et al. reported a superhydrophobic coating that can be used in aircraft anti-icing systems [108]. In this paper, EP resin was modified with hydroxyl-capped fluorosilanes and doped with fluoropolymer (FP) particles to obtain a superhydrophobic coating with integrated structure and excellent mechanochemical properties. As shown in Figure 14e, the freezing times of droplets on bare aluminum, FP-100% coating, and FP-200% coating were 14, 839, and 920 s, respectively. Qi et al. reported a superhydrophobic surface with anti-icing robustness [109]. The superhydrophobicity was enhanced using fluorine-modified titanium dioxide (TiO_2_) and fumed silica (SiO_2_) with a water contact angle of 161° and a freezing delay time of approximately 93 min. The modification process used KH550. As shown in Figure 14f, the water droplets on the unetched ASA surface are prone to freezing, showing a freezing delay time of about 7 min, and the surface with the best hydrophobicity possesses a freezing extension time of 103 min.

Silicone-based superhydrophobicity also has many advantages in achieving industrialization [110]. First, most of the reaction conditions of organosilicon are relatively gentle, and the reaction can be completed using an acidic and alkaline catalyst, without imposing high temperature and pressure, which makes it simple and fast in the preparation process and convenient for mass production [111]. Secondly, silicone-based superhydrophobicity is fast and easy to operate in the end-use process, which is conducive to commercialization. Third, silicones have good high-temperature resistance [112] and aging resistance [113]. The Si-O bond does not break or decompose even at high temperatures. So, industrialized products can be adapted to most environments in terms of temperature. The excellent performance of superhydrophobicity provides a vast application prospect in the fields of industry [114], agriculture, medicine [115], and daily life [116]. Yet, its low durability and high process expense greatly limit the industrial production of superhydrophobic materials on a large scale. Silicone superhydrophobic surfaces have improved the problems of easy aging and short lifetime, but have still suffered from the defects of low structural strength and mechanical stability. Some researchers have improved the durability of superhydrophobic materials somewhat [117], but their processes in production are complicated and costly. Therefore, addressing these two challenges is also the main challenge of our current scientific investigations. Solving these two problems is also the main research direction of our scientific work.

## 6. Conclusions and Outlook

A silicone-based superhydrophobic surface has been considered to be excellent in the application prospects, which should promote the practical usage of superhydrophobic materials. In this review, we introduced the polymerization of silicone monomers for materials, such as dehydration condensation and hydrosilation; it also summarized how silicones could play a role in the superhydrophobic preparation process, such as the lowering of surface energy via the grafting method, the building of a micro-/nano-structure via condensation cross-linking, and by acting as a binder in the reaction. It also summarizes the preparation methods of silicone superhydrophobic surfaces, such as electrostatic spinning, spraying, the template method, particle coating, the chemical deposition method, etc.; it also summarizes the application directions of silicone superhydrophobic materials such as self-cleaning, oil–water separation, anti-icing, and anti-corrosion. Organosilicon superhydrophobic materials have good prospects, but there are still some challenges to be solved.

(1) In the preparation of super-dual hydrophobic and superoleophobic surfaces, fluorosilanes are inevitably introduced, but fluorosilanes are very polluting to the environment, so it is a major challenge to realize the preparation of super-dual hydrophobic coatings without introducing fluorosilanes.

(2) Organosilanes form Si-O bonds when bonded to surface hydroxyl groups, but when the substrate material is a polymer such as plastic, the C-O-Si bonds formed are very unstable, so solving the problem of efficient bonding at the interface between organosilanes and polymers such as plastic is also a major challenge.

(3) Although most silicones are low toxicity and non-polluting, the use of the process will use a lot of organic solvents, and these alcohol organic solvents in the environment can be caused by pollution, which is also a problem that needs to be solved. Thus, in practical applications, aqueous solutions of silicones would be a good choice, but the inherent hydrophobicity of aqueous silicones again needs to be taken into account.

(4) Silicone-constructed superhydrophobic materials usually face the disadvantage of low mechanical stability. This drawback is a serious obstacle to the practical application of superhydrophobic surfaces.

Low durability is the biggest problem faced by superhydrophobic surfaces prepared by silicone. Superhydrophobic formation requires the combination of micro-/nano-structures and low surface energy. No material will have too much strength when it reaches the nanometer level, which is also the problem that many studies have tried to address. Some researchers prepare superhydrophobic coatings in very strong or thick porous structures so that the materials exposed after the upper layer of materials are destroyed and also have superhydrophobic properties. However, this strategy is not universal and must fundamentally solve the durability problem. Some work also builds micron-scale grids to protect the internal nano-structures, but these can only enhance the wear resistance and cannot resist physical impacts such as cutting and stabbing. Therefore, how to improve surface durability is a problem that all superhydrophobic materials need to solve, and it is also our research motivation and goal.

Although there are some challenges for silicone-based superhydrophobic materials, the advantages such as strong anti-aging properties, excellent heat resistance, simple and fast preparation methods, and low reaction conditions make them very valuable to both academic research and applications. The future investigation of silicone-based superhydrophobic materials can be focused on the durability of micro-/nano-structure, the functionality of interface, the cost-efficient preparation, etc. We envision that the silicone-based materials should be indispensable for the bioinspired superhydrophobic surface from fabrication to application.

We hope that this review will stimulate new thinking in silicone superhydrophobic surfaces, and will help others to understand the mechanism of preparation of silicone superhydrophobic surfaces and be aware of their progress and latest applications. This review can be enlightening for the research of polymer interface and colloid science.

## Figures and Tables

**Figure 1 polymers-15-00543-f001:**
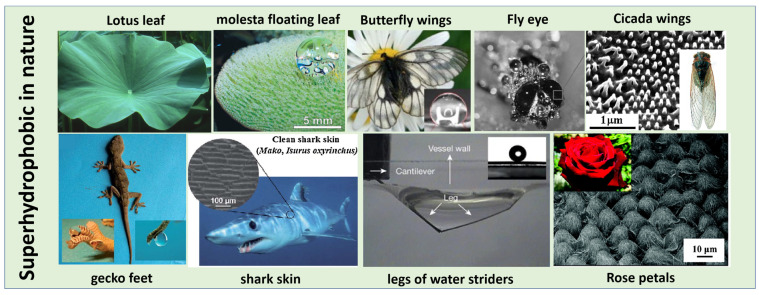
Superhydrophobicity in nature, including lotus leaves (Reprinted with permission from Ref. [2]. 2011, Beilstein J. Nanotechnol), morphology molesta floating leaves(Reprinted with permission from Ref. [3]. 2010, Adv. Mater), butterfly wings (Reprinted with permission from Ref. [4]. 2015, Mater. Today), fly eyes (Reprinted with permission from Ref. [5]. 2014, Small), cicada wings (Reprinted with permission from Ref. [6]. 2009, J. Exp. Biol), gecko feet (Reprinted with permission from Ref. [7]. 2012, Nanoscale), shark skin (Reprinted with permission from Ref. [8]. 2013, Adv. Funct. Mater) legs of water striders (Reprinted with permission from Ref. [9]. 2004, Nature) and rose petals (Reprinted with permission from Ref. [10]. 2008, Langmuir).

**Figure 2 polymers-15-00543-f002:**
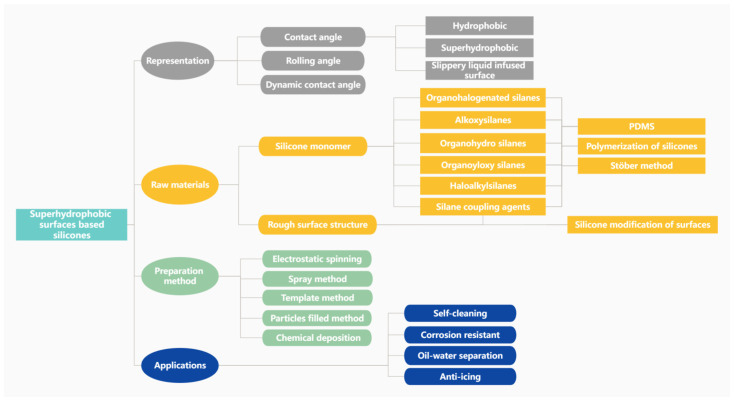
The contents of this paper’s review.

**Figure 5 polymers-15-00543-f005:**
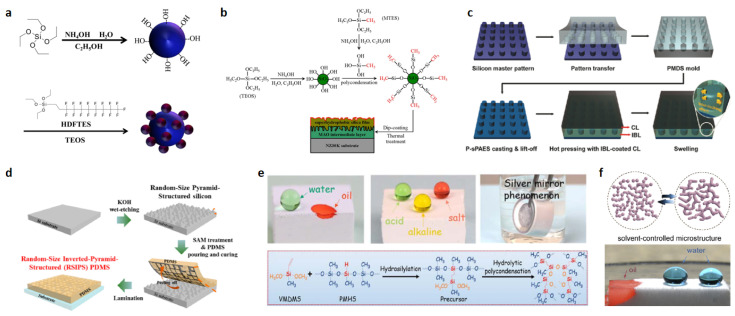
(**a**) Synthesis route of raspberry-like nanoparticles. Reprinted with permission from Ref. [42]. 2016,Ceram. Int. (**b**) Schematic illustration of superhydrophobic silica film on NZ30K magnesium alloy by combining MAO and sol–gel method. Reprinted with permission from Ref. [43]. 2012, Coat. Technol. (**c**) Schematic diagram of the fabrication of the pillar P-SPAES membrane and its working principle of interlocking effects (following the arrows from top left). Reprinted with permission from Ref. [44]. 2015, Adv. Mater. (**d**) Schematic of the soft-lithography process for random-size inverted-pyramid-structured polydimethylsiloxane sticker. Reprinted with permission from Ref. [45]. 2017, ACS Appl. Mater. Interfaces. (**e**) Preparation of the silicone precursor via a typical hydrosilylation reaction between the vinyl-methyl-dimethoxysilane and polymethylhydrosiloxane molecules under ambient temperature and the cross-linked silicone network via the hydrolytic polycondensation reaction among the methoxy groups of the precursor. Reprinted with permission from Ref. [46]. 2022, Mater. Today Chem. (**f**) Changing network of the silicone sponges by the type and concentration of co-solvents. Reprinted with permission from Ref. [47]. 2019, Colloid Interface Sci.

**Figure 7 polymers-15-00543-f007:**
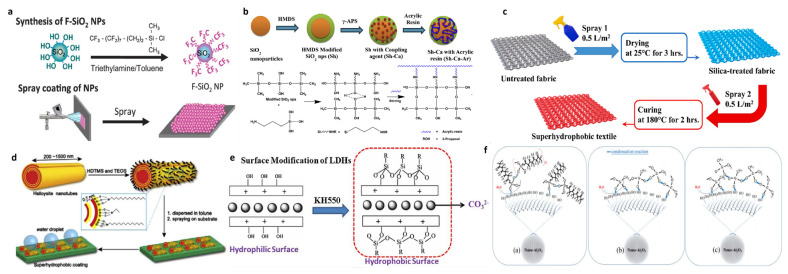
(**a**) Schematic diagram of the superhydrophobic layer of fluorinated silica nanoparticles synthesized by spray coating. Reprinted with permission from Ref. [59]. 2014, Chem. Comm. (**b**) Schematic diagram of superhydrophobic membrane formation and reaction process. Reprinted with permission from Ref. [60]. 2020, J. Clean. Prod. (**c**) Process for preparing superhydrophobicity via two-component spraying.Reprinted with permission from Ref. [61]. 2021, Prog. Org. Coat. (**d**) Schematic diagram of superhydrophobic POS@HNTs surface synthesis. Reprinted with permission from Ref. [62]. 2018, Appl. Chem. Eng. J. (**e**) Schematic diagram of superhydrophobic preparation using γ-methacryloxypropyltrimethoxysilane. Reprinted with permission from Ref. [63]. 2018, Appl. Surf. Sci. *Constr. Build Mater*. (**f**)
Schematic diagram of Al_2_O_3_ surface grafting of nano-Al_2_O_3_ (**a**) 1H,1H,2H,2H-Perfluorodecyltrimethoxysilane, (**b**) MTES and (**c**) MTES/ DMDES. Reprinted with permission from Ref. [64]. 2018, Prog. Org. Coat.

**Figure 8 polymers-15-00543-f008:**
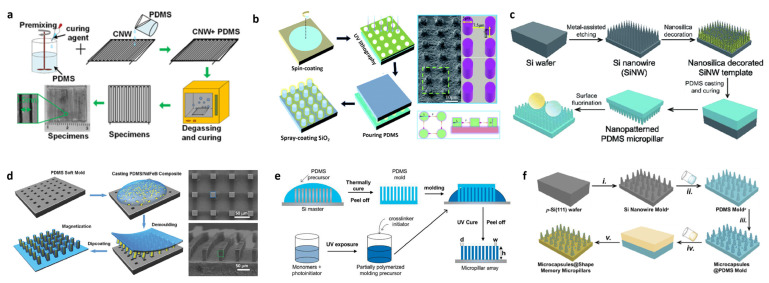
(**a**) Production process of CNW/PDMS bionanocomposite films. Reprinted with permission from Ref. [65]. 2020, Langmuir (**b**) Schematic diagram of the preparation process of PDMS. Reprinted with permission from Ref. [66]. 2018, *RSC Adv.* (**c**) Schematic diagram of a superhydrophobic PDMS micropillar surface prepared using SiNW decorated with silica nanoparticles as a template. Reprinted with permission from Ref. [67]. 2019, RSC Adv. (**d**) Schematic illustration of the fabrication of magnetic micropillar arrays. Reprinted with permission from Ref. [50]. 2018, Adv. Funct. Mater. (**e**) Schematic diagram of superhydrophobic hydrogel micropillar array fabricated using the template method. Reprinted with permission from Ref. [68]. 2010, Acc. Chem. Res. (**f**) Preparation of shape memory microcolumn surfaces using the template method. Reprinted with permission from Ref. [69]. 2020, ACS Appl. Mater. Interfaces.

**Figure 9 polymers-15-00543-f009:**
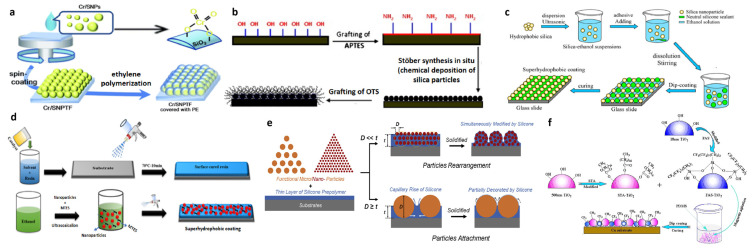
(**a**) Schematic diagram of the preparation of Cr/SNPTF superhydrophobic surface. Reprinted with permission from Ref. [70]. 2011, J. Mater. Chem. (**b**) Experimental strategy for hydrophobic substrate [(CH_2_)_18_]. Reprinted with permission from Ref. [71]. 2011, Colloids Surf. (**c**) Schematic procedure of superhydrophobic coating preparation. Reprinted with permission from Ref. [72]. 2019, Thin Solid Films. (**d**) Schematic of the preparation steps of the superhydrophobic coating. Reprinted with permission from Ref. [73]. 2022, Prog. Org. Coat. (**e**) Diagram of “glue + powder” method. Reprinted with permission from Ref. [24]. 2019, Matter. (**f**) Schematic diagram of the preparation process of superhydrophobic surface on Cu surface. Reprinted with permission from Ref. [74]. 2015, Colloids Surf.

**Figure 10 polymers-15-00543-f010:**
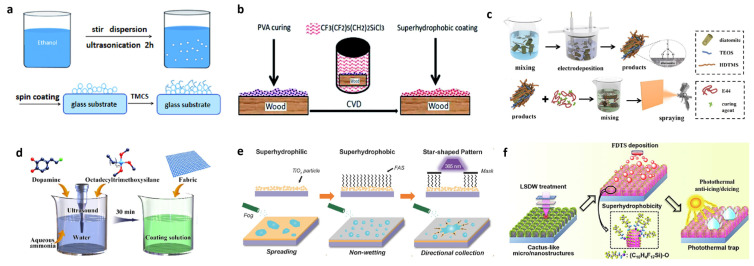
(**a**) Process flow for depositing superhydrophobic coatings on glass substrates. Reprinted with permission from Ref. [52]. 2017, RSC Adv. (**b**) Schematic diagram of vapor phase deposition. Reprinted with permission from Ref. [75]. 2017, RSC Adv. (**c**) Schematic diagram of the process of synthesizing superhydrophobic cetyltrimethoxysilane-modified diatomaceous earth powder and superhydrophobic composite layers via electrochemical deposition. Reprinted with permission from Ref. [76]. 2022, Prog. Org. Coat. (**d**) Schematic diagram of the synthesis of self-healing superhydrophobic fabrics. Reprinted with permission from Ref. [77]. 2020, Langmuir. (**e**) Preparation process of star-shaped wettability patterns. Reprinted with permission from Ref. [78]. 2014, Adv. Mater. (**f**) Schematic diagram of the preparation of a multifunctional aluminum sheet. Reprinted with permission from Ref. [79]. 2022, Chem. Eng. J.

**Figure 11 polymers-15-00543-f011:**
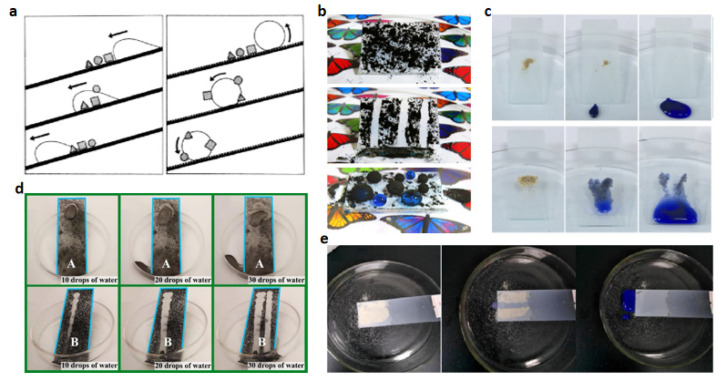
(**a**) Schematic diagram of the superhydrophobic self-cleaning principle. Reprinted with permission from Ref. [89]. 1997, Planta. (**b**) Self-cleaning properties of superhydrophobic coating. Reprinted with permission from Ref. [85]. 2017, Prog. Org. Coat. (**c**) Self-cleaning process of superhydrophobic coating and blank glass. Reprinted with permission from Ref. [90]. 2020, ACS Omega. (**d**) Self-cleaning pictures of sample A surface and self-cleaning pictures of sample B surfac. Reprinted with permission from Ref. [91]. (**e**) Colloid Interface Sci. Self-cleaning process of B3 coating. Reprinted with permission from Ref. [39]. 2019, Langmuir.

**Figure 12 polymers-15-00543-f012:**
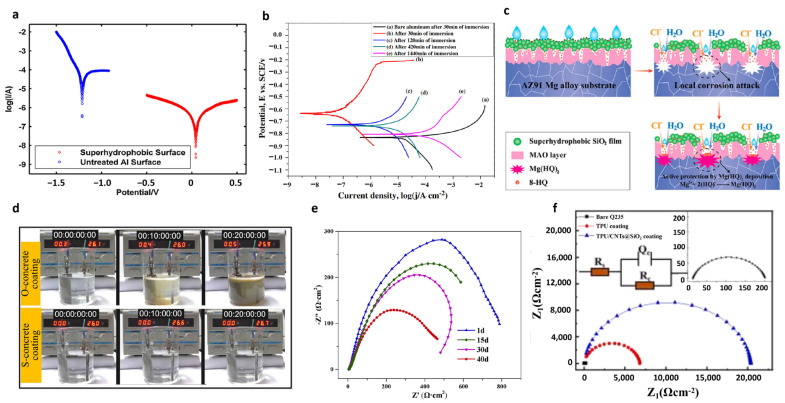
(**a**) Comparison of Tafel curves for untreated surfaces and superhydrophobic surfaces. Reprinted with permission from Ref. [92]. 2014, Appl. Surf. Sci. (**b**) Comparison of potential polarization curves of aluminum and superhydrophobic surfaces. Reprinted with permission from Ref. [93]. 2013, Appl. Surf. Sci. (**c**) Schematic diagram of the corrosion-resistance mechanism of the composite superhydrophobic coating. Reprinted with permission from Ref. [94]. 2017, Colloids Surf. (**d**) Electrochemical corrosion process and its results. Reprinted with permission from Ref. [95]. 2014, Surf. Coat. Technol. (**e**) Nyquist plots of different samples after exposure to 3.5 wt.% NaCl solution soaked at different times. Reprinted with permission from Ref. [96]. 2019, Ind. Eng. Chem. Res. (**f**) Nyquist plot (inset shows equivalent circuit). Reprinted with permission from Ref. [97]. 2019, Colloid Interface Sci.

**Figure 13 polymers-15-00543-f013:**
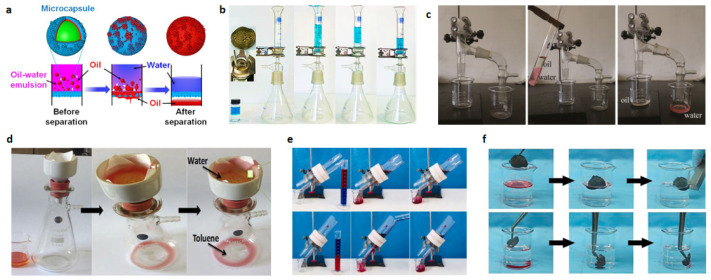
(**a**) Schematic diagram of oil–water separation principle. Reprinted with permission from Ref. [87]. 2020, ACS Appl. Mater. Interfaces. (**b**) Oil–water separation studies of the superhydrophobic and superoleophilic 3D graphene-based materials. Reprinted with permission from Ref. [100]. 2021, Sci. Total Environ. (**c**) Photographs of simple instruments fabricated by ourselves, before and after separation of the mixture of gasoline and colored water. Reprinted with permission from Ref. [101]. 2013, Appl. Surf. Sci. (**d**) Toluene/water separation process. Reprinted with permission from Ref. [60]. 2020, J. Clean. Prod. (**e**) Filtration separation processes for hexadecane/water and dichloromethane/water. Reprinted with permission from Ref. [102]. 2022, Hazard. Mater. (**f**) Demonstration of remote control absorption of light oil atop water and heavy oil underwater using modified silica sponges (MSSs). Reprinted with permission from Ref. [103]. 2018, Chem. Eng. J.

**Figure 14 polymers-15-00543-f014:**
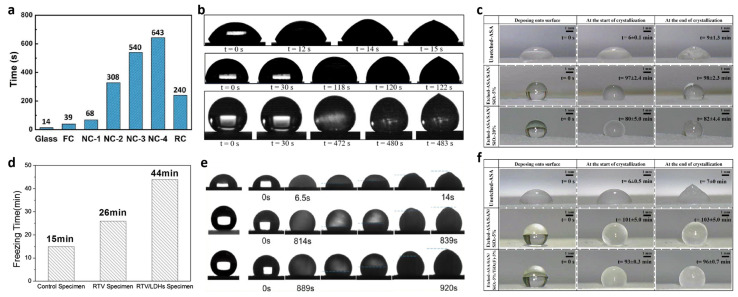
(**a**) Icing delay time of pristine and coated glass samples. Reprinted with permission from Ref. [106]. 2023, Chem. Eng. (**b**) The freezing time of water droplets on different samples at different sub-freezing temperatures and stages of water droplet freezing on aluminum, SILIKOPHEN AC1000, and SHP coating at −30 °C. Reprinted with permission from Ref. [73]. 2022, Prog. Org. Coat. (**c**) Freezing process of a water droplet of 10 μL on sample surface. During the measurement, the temperature was −10 °C and the humidity was 40%. Reprinted with permission from Ref. [107]. 2022, Appl. Surf. Sci. (**d**) Freezing times of water droplets on different asphalt mixture surfaces. Reprinted with permission from Ref. [63]. 2018, Constr. Build Mater. (**e**) Droplet icing process. Reprinted with permission from Ref. [108]. 2021, Colloids Surf. (**f**) Freezing process of a water droplet on sample surface. During the measurement, the temperature was −10 °C and the humidity was 40%. Reprinted with permission from Ref. [109]. 2021, Appl. Surf. Sci.

## Data Availability

Not applicable.

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
