# Peer review of "Advances in Bioinspired Superhydrophobic Surfaces Made from Silicones: Fabrication and Application"

_polymers, 2023, doi:10.3390/polym15030543_

Round 1

Reviewer 1 Report

This review introduces silicone-based superhydrophobic materials, in details, citing many papers on everything from their production methods to their functions. It is an important and significant review, but the authors should revise the following points. Minor revisions are recommended.

Major request for revision: 

There are a large number of abbreviations used without definitions. Abbreviations should be used after indicating the exact terms.

In many parts, symbols used in the original paper seem to be used as they are. This is also undesirable. Please use simplified substance names that allow us to imagine the contents, rather than meaningless abbreviations.

For instance, 

1) In 2.5 Definition of Superhydrophobicity, what is SLP?

SLIPS: Slippery Liquid-Infused Porous Surfaces?

2) In Figure 4, KH560 is what?

3) In 3.4.3, SCBS?

4) In lines 579 and 630,  polydimethylsiloxane (PDMS) are defined. However, PDMS is used repeatedly in the previous sentences.

Simple mistakes, probably.

1) Lines 239,262; the equation number 7 is used doubly.

2) Lines 660-66, two sentences with the same meaning overlap and are redundant.

3) Lines 775-811, Figure 12 seems to be labeled as "Figure 121".

4) In Line 792, "Forty grams of silicate cement"   

I don't understand why you bother describing 40g here.

5) In line 837, "it" might be "It".

6) In line 840, "Reported" might change to "reported".

7) In lines 1910-911, "As shown in Figure 14e. The freezing time" will be "As shown in Figure 14e, the freezing time".  

Reviewer 2 Report

A review of advances in bioinspired superhydrophobic surfaces made from silicones: fabrication and application: A few points deserve attention for further publication. I suggest that it is accepted for publication after the following revisions:

 - The authors could clarify in the abstract of the manuscript the mechanism, advantages, problems, and solutions for the superhydrophobic surfaces made from silicones systems.

 - In addition, authors should highlight the advantages / disadvantages of these superhydrophobic surfaces made from silicones methods for industrial application and how this information will be addressed in the manuscript.

 - Advantages for the superhydrophobic surfaces made from silicones: Which methods have advantages? Are they simple methods contribution? When compared with other sustainable techniques? Authors need to leave these clear information to throughout the text and the methods discussed in this manuscript. In addition, this information is needed for the superhydrophobic surfaces made from silicones systems contribution protocols are applied on an industrial scale.

 - Problems with the superhydrophobic surfaces made from silicones systems: Does this protocol have a significant problem? This discussion could be improved.

 - Additionally, advances in the superhydrophobic surfaces made from silicones systems with engineered tailor-made have been performed with other strategies. May open new opportunities. This discussion could be improved.

 - This review had broached interests in the progress and recent applications of the superhydrophobic surfaces made from silicones: The main contributions to the accomplishment of this work must be included in the conclusion.

 - Please, check all references according to the author's instructions.

- The manuscript must be formatted according to the journal's standards.
